# TiVA: Time-aligned Video-to-Audio Generation

Xihua Wang*
xihuaw@ruc.edu.cn
Gaoling School of Artificial
Intelligence, Renmin
University of China

Yuyue Wang*
wangyuyue123@ruc.edu.cn
Gaoling School of Artificial
Intelligence, Renmin
University of China

Yihan Wu*
yihanwu@ruc.edu.cn
Gaoling School of Artificial
Intelligence, Renmin
University of China

Ruihua Song†
songruihua_bloon@outlook.com
Gaoling School of Artificial
Intelligence, Renmin
University of China

Xu Tan†
xuta@microsoft.com
Microsoft

Zehua Chen
zhc23thuml@mail.-
tsinghua.edu.cn
Department of Computer
Science and Technology,
Tsinghua University

Hongteng Xu
hongtengxu313@gmail.com
Gaoling School of Artificial
Intelligence, Renmin
University of China

Guodong Sui
guodong.sui.001@gmail.com
ZHI-TECH GROUP

## Abstract

Video-to-audio generation is crucial for autonomous video editing and post-processing, which aims to generate high-quality audio for silent videos with semantic similarity and temporal synchronization. However, most existing methods mainly focus on matching the semantics of the visual and acoustic modalities while merely considering their temporal alignment in a coarse granularity, thus failing to achieve precise synchronization. In this study, we propose a novel time-aligned video-to-audio framework, called TiVA, to achieve semantic matching and temporal synchronization jointly when generating audio. Given a silent video, our method encodes its visual semantics and predicts an audio layout separately. Then, leveraging the semantic latent embeddings and the predicted audio layout as condition, it learns a latent diffusion-based audio generator. Comprehensive objective and subjective experiments demonstrate that our method consistently outperforms state-of-the-art methods on semantic matching and temporal synchronization.

## CCS Concepts

• **Information systems** → **Multimedia content creation**; • **Applied computing** → Sound and music computing.

## Keywords

Audio Generation, Video Processing, Diffusion Models, Controllable Generation, Multimedia Generation

**ACM Reference Format:**
Xihua Wang, Yuyue Wang, Yihan Wu, Ruihua Song, Xu Tan, Zehua Chen, Hongteng Xu, and Guodong Sui. 2024. TiVA: Time-aligned Video-to-Audio Generation. In *Proceedings of the 32nd ACM International Conference on Multimedia (MM '24), October 28–November 1, 2024, Melbourne, VIC, Australia.* ACM, New York, NY, USA, 10 pages. https://doi.org/10.1145/3664647.3681027

---

*Equal contributions.
†Corresponding authors.

---

## 1 Introduction

Video-to-audio (V2A) generation aims to generate synchronized and realistic audio solely based on silent videos [1]. This task is significant for many practical applications, including autonomous video editing and post-processing. Especially, achieving high-quality V2A is necessary for the current AI-generated videos because of their silent nature, e.g., Sora generated videos [4].

As a main challenge of V2A generation, given a silent video, we need to not only ensure the semantics of the generated audio to match well with the visual content but also make the audio temporally synchronize with the video. However, most existing V2A generation methods mainly focus on matching the semantics of the visual and acoustic modalities while seldom considering their temporal alignment. In particular, they often follow the learning paradigms of existing text-to-audio methods [14, 15, 21], applying contrastive learning-based models to encode video/images as conditions and then generating Mel-spectrograms [25] or audio tokens [16, 29] via diffusion or auto-regressive models, respectively. Some recent methods, like DiffFoley [25], maximize the similarity of audio-visual pairs from the same time segment and minimize the similarity of those across different time segments. However, this temporal contrastive learning only aligns the generated audio with the input video in a coarse granularity, failing to achieve precise temporal synchronization.

In this study, we propose a novel time-aligned V2A generation method, called TiVA, to achieve semantic matching and precise temporal synchronization jointly when generating audio. In principle, our method is based on the fact that the low-resolution Mel-spectrogram of an audio provides an effective audio layout for temporal synchronization. Take Figure 1 as an example. Given a video of cat, in which a cat opens its mouth wide when hissing and more narrowly when meowing, both the ground-truth waveform and its corresponding Mel-spectrogram are highly synchronized with the dynamics of the visual content. By reducing the resolution of the Mel-spectrogram, we obtain an audio layout, as shown in Figure 1 (d). On one hand the layout provides precise alignment

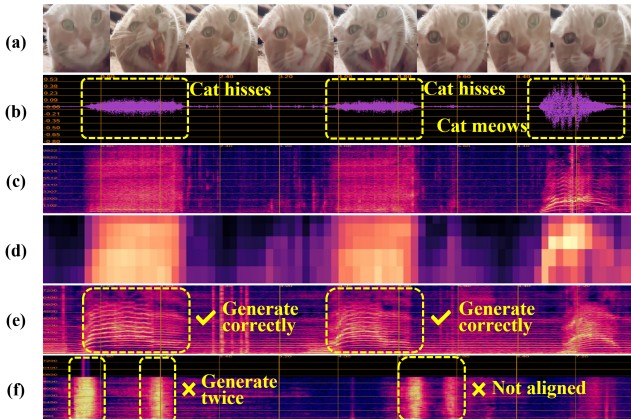

**Figure 1: An example of video-to-audio generation. (a) shows frames of a video in which a cat hisses twice and meows once; (b) shows the waveform of the video's corresponding audio; (c) shows the Mel-spectrogram of the audio; (d) shows an ideal audio layout provided by low-resolution Mel-spectrogram; (e) shows the Mel-spectrogram of our generated audio; (f) shows the Mel-spectrogram generated by DiffFoley [25].**

between the mouth opening and making sounds three times; on the other hand the frequency difference of hissing and meowing can also be easily distinguished by the shapes of bright areas in the layout. Thus the audio layout can be used to supervise the learning of the proposed model. In particular, in the training phase, we learn a conditional diffusion-based audio generator and an associated audio layout predictor jointly. The audio layout predictor predicts the low-resolution Mel-spectrogram based on the input video. Taking the real audio layout and the semantical encoding result of the video as conditions, the audio generator generates a prior of the audio in a latent space. In the testing phase, the predictor estimates the audio layout, which provides useful temporal condition for the learned audio generator. As shown in Figure 1 (e, f), compared to the state-of-the-art V2A method [25], our TiVA can generate an audio from the input video, with better temporal synchronization.

There are three main contributions of our work:

- We propose using low-resolution Mel-spectrograms of ground-truth audio to self-supervise the prediction of an audio layout, which bridges temporal encoding and audio generation. Experimental results indicate that it can well capture temporal alignment information and be complementary to semantic information in controlling audio generation.
- We propose a new framework called TiVA that jointly learns two separated encoders to embed semantic and temporal information and a conditional diffusion-based audio generator. Experimental results show that TiVA is effective to improve both semantic matching and precise temporal synchronization while accelerating the generation speed by about 40%.
- We perform comprehensive evaluations of all methods through extensive experiments, assessing generation quality, temporal synchronization, and semantic and efficiency metrics on the public in-domain dataset. Additionally, we test our proposed method

on out-of-domain Sora-generated videos [1], yielding promising outcomes. Demonstration samples can be accessed at https://tiva 2024.github.io/TiVA.github.io/.

## 2 Related Work

### 2.1 Condition Encoding for Audio Generation

Audio generation exploits various condition information: (1) Sole semantic condition, typically extracted using pre-trained models (e.g., FLAN-T5 [6], ResNet [11], VideoMAE [30], CLIP [27], CLAP [31]) for semantic guidance in audio generation [15, 20, 21, 26, 32]. (2) Enriched semantic condition. Make-an-Audio-2 [14] and WavJourney [23] leverage large language models to augment descriptions of audio, while DiffFoley [25] injects temporal dynamics into semantic condition through advanced temporal contrastive learning. (3) Additional information condition. Beyond semantics, SpecVQGAN [16] and SyncFusion [7] improve audio quality by employing additional video flow and onset information respectively. Different from them, our work proposes a novel audio layout condition, which captures temporal alignment information and complements semantic information in controlling audio generation.

### 2.2 Generative Modeling for Audio Generation

With encoded conditions from input modalities, previous studies on audio generation generally fall into two streams: (1) Autoregressive methods, like SpecVQGAN [16], AudioGen [20], and IM2WAV [29], employ transformer-based architectures for sequential audio generation, yielding high-quality output but at the expense of computational time. (2) Non-autoregressive methods, like SoundStorm [3] and MagNet [34], adopt multi-step non-autoregressive designs for more efficient generation, but with lesser audio quality compared to autoregressive methods. Recent advancements incorporate diffusion models, with DiffSound [32] adopting discrete diffusion and works like DiffFoley [25] and the AudioLDM [21, 22] and Make-an-audio [14, 15] series employing latent diffusion models (LDM), achieving enhanced audio quality and generation speed through well-designed samplers [24]. Our study further advances LDMs by integrating both semantic embeddings and a novel audio layout as conditions. We also introduce effective strategies for optimizing the two condition encoders/predictors and the audio generator.

### 2.3 Automatic Evaluation of Audio Generation

Previous research [16, 21, 29, 32] assesses the *quality* of generated audio by comparing it with the ground truth using automatic metrics such as Fréchet Inception Distance (FID) [16], mean KL divergence (MKL) [16], and Inception Score (IS) [16], derived from feature distributions of audio classifiers (Melception[16], PANNs [19], VGGish [12]). FID measures dataset-level consistency, MKL quantifies pair-level variation, and IS estimates diversity via audio distribution entropy. Moreover, FoleyGen [26] employs ImageBind score (IB) to measure semantic *relevance* between generated audio and input video based on features extracted by the pre-trained multi-modal alignment model ImageBind [10]. Recent studies have extended the evaluation to measure the *synchronization* between generated audio and input video. For instance, DiffFoley [25] applies a binary

[1] https://openai.com/research/video-generation-models-as-world-simulators

classifier, while CondFoleyGen [8] employs a multi-class classifier [17] to predict the temporal offset (TO) between an audio-video pair. SyncFusion [7] implements synthesized onset accuracy (Onset Acc) and average precision score (Onset Sync AP) via predicting binary onset signal. However, these onset signal-based metrics may not adequately capture the fine-grained details across the entire temporal distribution. To address this limitation, we propose two fine-grained synchronization evaluation metrics based on the more detailed onset detection function (ODF) curves in this paper.

## 3 Method

Video-to-audio (V2A) generation aims to synthesize an audio track $a$ from a silent video $v$ comprising $N$ frames $v : [f_1, ..., f_i, ..., f_N]$. Ideally, the generated audio $a$ should precisely align with the visual content of $v$, both semantically and temporally. Previous works have achieved coarse alignment between the generated audio and input visual information primarily through semantic guidance and control, e.g., accurately generating multiple cat sounds rather than a single dog sound for the video depicted in Figure 1 (a, f). However, it remains a challenge for these methods to achieve a fine-grained alignment, which includes more intricate semantic sound structures such as the distinction between different sounds produced by the same entity, e.g., a cat's hiss versus its meow, and precise temporal information, e.g., the exact start and end time of a sound.

To address the above challenge of precise alignment, we adopt a coarse-to-fine principle. Given the difficulty in directly achieving precise alignment in one step, as evidenced by previous work, our approach initially predicts a simpler intermediate state (low-resolution Mel-spectrograms in our method) before generating the final output (high-resolution raw Mel-spectrograms). We propose defining this low-resolution Mel-spectrogram as a new control signal, termed *audio layout*, serving as the bridge in video-to-audio generation, as exemplified in Figure 1 (d) and Figure 2 (e). Then, we propose a new V2A framework, named TiVA, which first produces semantic embeddings and this audio layout, and then utilizes both as condition to control the audio generation, as shown in Figure 3 (a). We will introduce the audio layout, the architecture of TiVA, and the training of TiVA in detail in the following subsections.

### 3.1 Audio Layout

In the domain of 2D image generation, precise control is typically achieved by 2D layout inputs [33]. As audio is often represented in a 2D Mel-spectrogram format (with time and frequency as its axes), akin to images, we propose the use of a coarse-grained Mel-spectrogram as an *audio layout* in the audio domain, denoted as $t$. This proposed audio layout is simpler for models to predict compared to raw Mel-spectrograms. Additionally, it can provide more detailed control signals, such as the sound structure and the start and end of a sound, than the Onset signals or ODF curves (extracted from audio waveforms), as illustrated in Figure 2. Experimental results, detailed in Section 4.3, confirm the superior effectiveness of the proposed audio layout in Mel-spectrograms over raw Mel-spectrograms target or Onset and ODF forms for V2A tasks.

Specifically, a ground truth audio layout $t_a$ is generated by downscaling a Mel-spectrogram of dimensions $[M^{\text{raw}}, C^{\text{raw}}]$ to a target resolution $[M, C]$. Here, $M^{\text{raw}}$ and $C^{\text{raw}}$ denote the original number

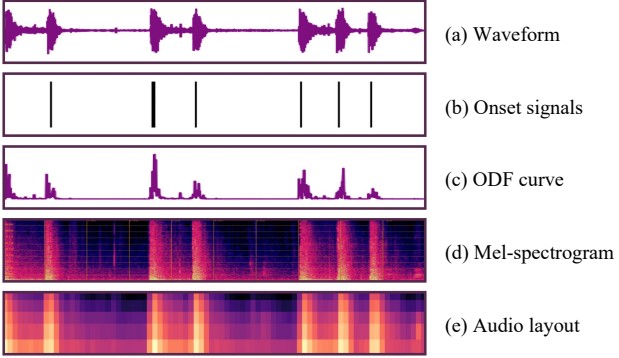

**Figure 2: Various representations of audio signals.**

of frames and frequency dimensions, respectively, while $M$ and $C$ denote the reduced temporal length and frequency dimensions:

$$t_a = \text{Resize}(\text{Norm}(\text{mel}_a); M, C). \tag{1}$$

This simple but efficient extraction facilitates a self-supervised learning manner, that captures the temporal structure pattern without requiring additional audio annotations. The extracted audio layout in size $[M, C]$ is further resized to match the latent space dimensions $[h, w]$ for integration into the latent diffusion model.

### 3.2 Architecture of TiVA

TiVA integrates a semantic encoder $\mathcal{S}$, a layout predictor $\mathcal{T}$, and a conditional generator $\mathcal{G}$. As illustrated in Figure 3 (a), TiVA extracts semantic embeddings $s$ and predicts an audio layout $t_v$ from the input video frames $[f_1, ..., f_i, ..., f_N]$ through $\mathcal{S}$ and $\mathcal{T}$, respectively. These conditions are then employed by $\mathcal{G}$ to guide the generation of an audio within a latent space. The following contents delineate the encoding and generative processes of TiVA.

*3.2.1 **TiVA's Condition Encoders**.* The semantic encoder and layout predictor transform the video frames into the semantic embeddings $s$ and the temporal audio layouts $t_v$. This begins with the extraction of multi-layer features $c_i^l$ via CLIP [27], where $c_i^l$ denotes the CLS and patch features from layer $l$ of CLIP for frame $f_i$. These features are then utilized to produce the corresponding semantic and temporal conditions $s$ and $t_v$, as illustrated in Figure 3 (b).
*Semantic Encoder $\mathcal{S}$* employs a linear layer to map the CLS features of CLIP's 11th layer into semantic embeddings for each frame:

$$s_i = \text{Linear}(\text{CLIP}(f_i)). \tag{2}$$

The $s_i$ from each frame collectively form the semantic condition $s$.
*Layout Predictor $\mathcal{T}$* employs an encoder-decoder structure. The encoder first concatenates and pools patch features from selected CLIP layers for each frame, serving as encoder inputs, denoted as $c_i^{\text{patch}}$. These features are then processed through windowed self-attention layers, facilitating interactions among adjacent frames:

$$
\begin{aligned}
c_i^{\text{patch}} &= \text{Pooling}(\text{Concatenate}([c_i^{4,\text{patch}}, c_i^{8,\text{patch}}, c_i^{11,\text{patch}}])), \\
c_i^{\text{patch}'} &= \text{Window-Encoder}(c_i^{\text{patch}}).
\end{aligned}
\tag{3}
$$

The decoder starts by combining CLS features and learnable query tokens $q_i$ as inputs $c_i^{\text{query}}$, and is followed by multiple decoder

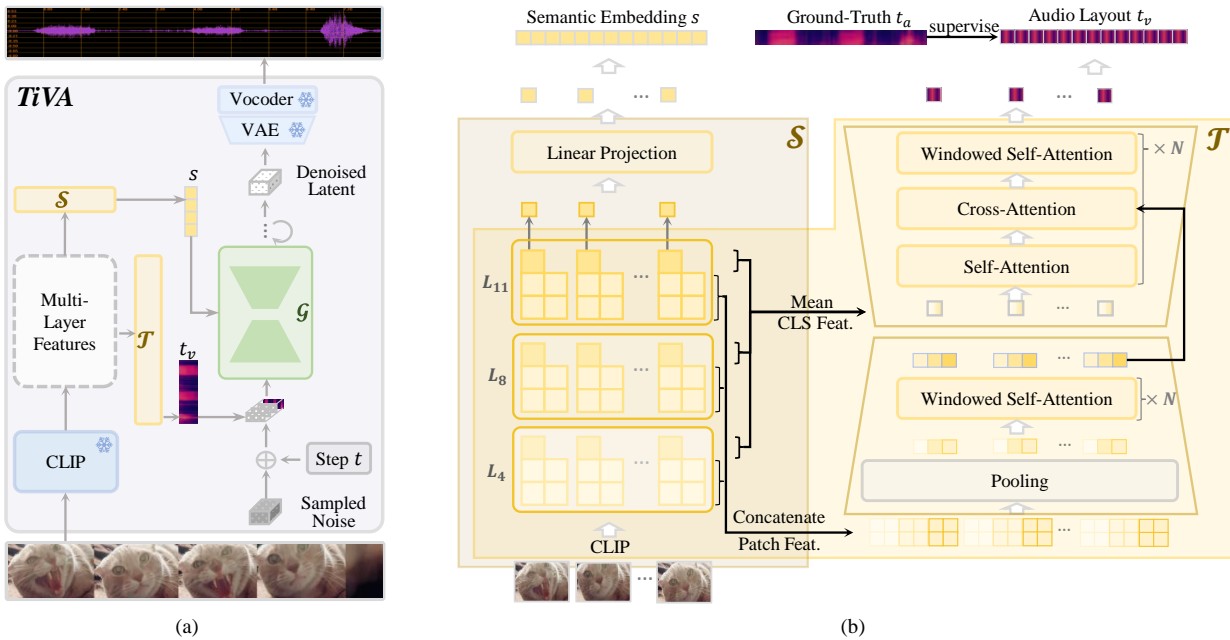

Figure 3: The architecture of our proposed Time-aligned Video-to-Audio Generation (TiVA) framework. (a) shows the overall architecture of TiVA. TiVA uses two modules to encode video frames: semantic encoder $\mathcal{S}$, which extracts semantic information, and audio layout predictor $\mathcal{T}$, which predicts temporal information. Then semantic and layout controlled generator $\mathcal{G}$ orchestrates a diffusion reverse process, synthesizing the outputs conditioned by both semantic and temporal control signals. (b) shows the detailed architecture of semantic encoder $\mathcal{S}$ and audio layout predictor $\mathcal{T}$, and how they process video frames.

blocks and an output layer. Each decoder block contains a self-attention layer, a cross-attention layer ($c_i^{\text{query}}$ as query, $c_i^{\text{patch}'}$ as key and value), and a windowed self-attention layer. The final output layer contains a single linear transformation, which maps the last decoder block's output $c_i^{\text{out}}$ to match the dimensions of the intended audio layout $t$, producing the predicted audio layout $t_v$:

$$c_i^{\text{query}} = \text{Pooling}(\text{Concatenate}([c_i^{4,\text{CLS}}, c_i^{8,\text{CLS}}, c_i^{11,\text{CLS}}])) + q_i,$$

$$c_i^{\text{out}} = \text{Window-Cross-Decoder}(c_i^{\text{query}}, c_i^{\text{patch}'}), \quad (4)$$

$$t_v = \text{Linear}(\text{Concatenate}([c_0^{\text{out}}, \ldots, c_N^{\text{out}}])).$$

### 3.2.2 TiVA's Conditional Generator.
TiVA's generator extends the Latent Diffusion Model (LDM) [28] for V2A generation. The LDM, using a Variational Autoencoder (VAE), converts audio Mel-spectrograms into a compact latent representation $z_a$ for efficient generation. Typical conditioned LDMs focus on generating latents $z_a$ given noisy latents and semantic embeddings $s$. During LDM training, the diffusion forward process first transforms the original ground-truth distribution $z_a$ into a standard Gaussian distribution by progressively adding Gaussian noise $\epsilon$ according to a pre-defined schedule $\alpha_1, \ldots, \alpha_T$. Here, $T$ denotes the total training time steps. The perturbed data $z_a^t$ at step $t$ is derived from the latents $z_a$ ($z_a^0$):

$$p(z_a^t | z_a^0) = \mathcal{N}(z_a^t; \sqrt{\bar{\alpha}_t} z_a^0, (1 - \bar{\alpha}_t)I), \quad (5)$$

where $\bar{\alpha}_t = \prod_{i=1}^t \alpha_i$. Consequently, the LDM's diffusion model, denoted as $\mathcal{G}$, is trained to predict the noise at each time step $t$:

$$\hat{\epsilon}^t = \mathcal{G}(z_a^t, s, t). \quad (6)$$

Semantic and Layout Controlled Generator $\mathcal{G}$. TiVA's generator $\mathcal{G}$ extends typical LDMs by integrating the audio layout $t$ with the semantic embedding $s$ as conditions. The latents $z_a$ are encoded from the Mel-spectrogram, typically via a convolutional VAE which preserves spatial relationships. The audio layout $t$, derived from the same Mel-spectrogram, inherently aligns with $z_a$ spatially. Thus, we append the resized $t \in [1, h, w]$ to $z_a \in [c, h, w]$ as an extra channel, where $c$, $h$ and $w$ denote the latent space dimensions. The appended latents, $\tilde{z}_a = [z_a, t] \in [(c+1), h, w]$, serve as the input to $\mathcal{G}$ at each diffusion step $t$, facilitating audio layout guided generation:

$$\hat{\epsilon}^t = \mathcal{G}([z_a^t, t], s, t). \quad (7)$$

Our generator $\mathcal{G}$ utilizes a Transformer-UNet (T-UNet) structure [22], integrating convolutional, self-attention, and cross-attention layers. The appended latents $\tilde{z}_a^t$ are fed into the T-UNet as input, with the semantic embedding $s$ incorporated into the cross-attention layers. It's noted that we only take the first $C$ dimensions from the $C + 1$ dimensions of $\mathcal{G}$'s direct output as estimated noise $\hat{\epsilon}^t$ for each step.

## 3.3 Training Strategies
In the training phase, given a video with sound, comprising silent frames $v : [f_1, \ldots, f_i, \ldots, f_N]$ and the ground-truth audio $a$, we separately optimize the layout predictor $\mathcal{T}$, the semantic encoder $\mathcal{S}$ and the generator $\mathcal{G}$, before a joint optimization stage.

The layout predictor is trained to predict the audio layout $t_v$ from video frames, using features from CLIP as input. This training is guided by the ground-truth audio layout $t_a$, derived from the

audio $a$'s Mel-spectrogram, and and aims to minimize the L2 loss:

$$\mathcal{L}_{\mathcal{T}} = \|\boldsymbol{t}_a - \boldsymbol{t}_v\|_2^2. \tag{8}$$

The semantic encoder $\mathcal{S}$ processes frames to produce the semantic embedding $\boldsymbol{s}$, which, along with the ground-truth $\boldsymbol{t}_a$, serves as conditions for the generator $\mathcal{G}$. The generator $\mathcal{G}$ predicts the noise sequence to reverse the diffusion process, iteratively denoising $z_a^t$ to reconstruct $z_a^0$. The optimization of $\mathcal{S}$ and $\mathcal{G}$ is driven by an L2 loss on top of $\mathcal{G}$'s output, targeting the noise prediction:

$$\mathcal{L}_{\mathcal{G}} = \|\epsilon^t - \mathcal{G}([z_a^t, \boldsymbol{t}_a], \mathcal{S}(v), t)\|_2^2, \tag{9}$$

where during training, 10% of the semantic embeddings $\boldsymbol{s}$ are randomly zeroed to promote unconditional generation capabilities.

Independent optimization of $\mathcal{T}$ and $\mathcal{G}$ leads to suboptimal performance. This is because $\mathcal{G}$, trained on the precise ground-truth $\boldsymbol{t}_a$ is then applied with the predicted $\boldsymbol{t}_v$ during inference. Due to $\mathcal{G}$'s acute sensitivity to deviations, it tends to magnify even minor discrepancies between $\boldsymbol{t}_a$ and $\boldsymbol{t}_v$. Empirically, we find this gap significantly impacts generation quality. To address this, we introduce a joint optimization stage in addition to the initial separate optimization stage. $\mathcal{T}$ first predicts $\boldsymbol{t}_v$ from the video, and then $\mathcal{G}$ takes it as input for the audio layout generation. The learning objective is to optimize the L2 loss on $\mathcal{G}$'s prediction to achieve a joint optimization of all components $\mathcal{G}$, $\mathcal{T}$, and $\mathcal{S}$:

$$\mathcal{L}_{\text{Joint}} = \|\epsilon^t - \mathcal{G}([z_a^t, \mathcal{T}(v)], \mathcal{S}(v), t)\|_2^2, \tag{10}$$

where 10% of $\boldsymbol{s}$, i.e., $\mathcal{S}(v)$, is set to zero in this joint optimization.

# 4 Experiments

## 4.1 Experimental Setup

*4.1.1 Datasets and Implementation Details.* We adopt the AudioSet-V2A dataset [9, 25], which includes ~400K filtered videos for training, and the VGGSound [5] dataset, which includes ~200K videos split by original VGGSound splits, following the same dataset setting in DiffFoley [25]. At the separated optimization stage, our layout predictor is trained on both AudioSet-V2A and the VGGSound training set, whereas the semantic encoder and generator are trained exclusively on the VGGSound training set. At the joint optimization stage, all modules are trained on VGGSound training set. We conduct evaluation on the VGGSound test set to maintain consistency with previous works and ensure fair comparisons.

Video frames and audio are sampled at 10 FPS and 16 kHz, respectively, followed by preprocessing according to the protocols of CLIP [27] and AudioLDM2 [22]. The size of the preprocessed Mel-spectrogram is [1024, 64], and the size of our audio layout $\boldsymbol{t}_a \in \mathbb{R}^{M \times C}$ is set to [102, 5]. During the training stage, the CLIP, the VAE and vocoder (sourced from AudioLDM2 [22]) are frozen. The training of latent diffusion model is governed by a variance-preserving DDPM scheduler [13], with the semantic condition $\boldsymbol{s}$ being dropped at a probability of 0.1. For inference, we employ the second-order DPM-Solver sampler [24] and apply a guidance scale of 5.0 to steer the generation. We set the inference step as 25.

*4.1.2 Baselines.* We collect or implement the following baselines that can address the V2A task on VGGSound dataset:

- **SpecVQGAN** [16] integrates a visual feature extractor, a VQ-VAE, an autoregressive transformer, and a vocoder. It utilizes

RGB+Flow and ResNet50 visual features, with the former outperforming. We employ the RGB+Flow configuration of the public code and model to generate 10-second audio samples.
- **DiffSound-V** adapts the DiffSound [32] architecture, replacing textual features with SpecVQGAN's RGB+Flow visual features, and initializing parameters to the released weights. It is tuned on the VGGSound training set and produces 10-second audio clips.
- **DiffFoley** [25] is a diffusion-based model that generates audio in latent space derived from Mel-spectrograms, conditioned on CAVP features. It generates 8-second clips using the official codes.
- **IM2WAV** [29] employs a two-stage autoregressive process, first generating discrete audio tokens given CLIP features, then refining it into high-fidelity audio. We generate 4-second audio clips with the public code and model.
- **FoleyGen** [26] employs a transformer to generate neural codecs autoregressively, conditioned on CLIP features. Due to the unavailability of its code, we compare its reported results with ours.

For consistency with DiffFoley, we use the first 8s from SpecVQGAN, DiffSound-V, and TiVA outputs. For IM2WAV, we concatenate two 4s samples corresponding to the 0-4s and 4-8s clips of the video.

*4.1.3 Evaluation metrics.* For objective evaluation, we use the following groups of metrics:

- **Quality metrics.** We employ automatic metrics Fréchet Audio Distance (FAD) [18], Inception Score (IS), and KL-divergence (KL), each with a classifier name for specificity. The practice is in line with previous work [16, 21, 25, 29].
- **Synchronization metrics.** Previous metrics like Onset Acc and Onset Sync AP rely on binary onset. However, as depicted in Figure 2, Onset is a sparse, discrete signal. To achieve a more fine-grained evaluation of synchronization, we adopt Onset Detection Function (ODF) [2] curves, which capture sound energy variations over time. Audio-video synchronization is assessed through a comparison of ODF curves of generated and reference audio. We apply the Dynamic Time Warping Distance[2] (*DTW-dis*) and Wasserstein Distance[3] (*W-dis*) to measure their distance:
  (a) **DTW-dis** first employs a dynamic programming algorithm to optimally align two time sequences, allowing the one-to-many cases. Subsequently, it distorts the two sequences according to the alignment and calculates point-by-point Euclidean distance.
  (b) **W-dis** regards two distributions $P$ and $Q$ as a series of mounds and evaluates the minimum cost to transform $P$ into $Q$, considering both the distance between mounds and the mound mass. In our evaluation, DTW-dis and W-dis view ODFs as time series and one-dimensional distributions, respectively. Lower distance implies better synchronization. Supplementary analysis reveals that DTW-dis and W-dis correlate well with human judgment, outperforming previous metrics: Onset Acc [7], Onset Sync AP [7], Align Acc [25], and Temporal Offset (TO) [8].
- **Semantic metrics.** We use ImageBind score (IB) [10] to assess semantic relevance between input videos and generated audios.
- **Efficiency metrics.** We computed the average inference time for each sample across ten batches, each containing 100 samples, excluding the time attributed to the vocoder module.

---

[2]https://en.wikipedia.org/wiki/Dynamic_time_warping
[3]https://en.wikipedia.org/wiki/Wasserstein_metric

**Table 1: Comparison of our method with baselines on the VGGSound test set. For comparability with prior research, we assessed multiple classifier versions of each quality metric ('*mel.*' denotes the Melception version, '*vgg.*' denotes the VGGish version, and '*pas.*' denotes the Passt version. Please refer to Section 2.3 for details). For those metrics not reported, we employ their publicly available codes to replicate and evaluate results, marking our replicated results with parentheses '()'; otherwise, a dash '-' signifies unavailable data or absent public codes. The highest performing score for each metric is in bold, while the second highest score is in underlined. CG and CFG here represent classifier guidance and classifier-free guidance respectively.**

| Method | Vis. Feat. | Guidance | Quality | | | | | Synchronization | | Semantic | Time |
| --- | --- | --- | --- | --- | --- | --- | --- | --- | --- | --- | --- |
| | | | FAD ↓ | | KL ↓ | | IS ↑ | W-dis ↓ | DTW-dis ↓ | IB (%) ↑ | (Infer.) |
| | | | *mel.* | *vgg.* | *mel.* | *pas.* | *mel.* | | | | |
| *AutoRegressive* | | | | | | | | | | | |
| SpecVQGAN[16] | RGB + Flow | - | 8.93 | (8.58) | 6.93 | (4.18) | 30.0 | 5.70 | 3.00 | 9.4 | 3.40s |
| IM2WAV[29] | CLIP | CFG | 11.40 | 6.41 | **5.20** | 2.54 | 39.3 | 4.35 | **2.59** | 19.0 | 3.53s |
| FoleyGen[26] | CLIP | CFG | - | 1.65 | - | 2.35 | - | - | - | 26.1 | - |
| *Diffusion* | | | | | | | | | | | |
| DiffSound[32]-V | RGB + Flow | CFG | 12.27 | 3.36 | 6.43 | 2.99 | 28.8 | 4.97 | 2.65 | 15.7 | 0.53s |
| DiffFoley[25] | CAVP | CFG | 11.20 | - | 6.36 | - | 53.3 | - | - | - | - |
| DiffFoley[25] | CAVP | CFG+CG | 9.87 | (4.89) | 6.43 | (3.10) | 62.4 | 4.21 | 2.59 | 18.5 | 0.25s |
| TiVA | CLIP | CFG | **8.71** | **0.88** | 5.98 | 2.12 | **64.9** | **3.85** | 2.59 | **31.0** | **0.15s** |

## 4.2 Comparing TiVA with Baselines

We benchmark our proposed TiVA against existing V2A methods on VGGSound dataset, following established protocols. As shown in Table 1, TiVA outperforms existing methods, achieving leading scores in FAD, KL (pas.), IS, W-dis, DTW-dis, and IB metrics, and ranks second in KL (mel.). Notably, TiVA's design enhances not only temporal synchronization but also generation quality and semantic relevance. This is caused by that the 2D audio layout provides a coarse prior of target audio that helps the generation of relevant sounds. Moreover, the results indicate that TiVA has a substantial leap in computational efficiency, accelerating audio generation by approximately 40% compared to baselines. This may be attributed to the well-designed architecture, in particular the sharing of CLIP features between the semantic encoder and audio layout predictor.

For subjective assessment, we ramdonly select 50 videos from VGGSound test set. Human evaluators are asked to rate audio samples from different models from four perspectives: Overall quality (Overall), Sound Quality (SoundQua), Semantic Relevance (SemRel), and Synchronization Score (SyncScore) on a 5-level Likert scale. Each model's audio output is rated by ten evaluators, and their scores are averaged and assessed for a 95% confidence interval. According to these subjective ratings presented in Table 2, TiVA consistently outperforms all baselines by a substantial margin. The IM2WAV model ranked the second, surpassing the other two baselines, SpecVQGAN and DiffFoley, in every evaluative criterion [4].

For qualitative evaluation, Figure 4 presents a series of cases in the in-domain VGGSound test set. Case 1 and 3 show that the audio generated by our TiVA model can better temporally align with the visual content or the ground-truth audio than baselines. In Case 1, DiffFoley exhibits delays for the second cat's hiss and the final meow, whereas IM2WAV delays the last meow. Both of them two miss certain sounds in Case 3. Case 2 and Case 3 demonstrate

---

[4]Visit our demo page for playable examples: https://tiva2024.github.io/TiVA.github.io/

**Table 2: Subjective evaluation results for different models. A rating ranges from 0 to 5, the higher the better.**

| Method | Overall ↑ | SoundQua ↑ | SemRel ↑ | SyncScore ↑ |
| --- | --- | --- | --- | --- |
| SpecVQGAN[16] | 1.10 ± 0.09 | 1.32 ± 0.10 | 1.14 ± 0.12 | 1.09 ± 0.12 |
| DiffSound[32]-V | 2.01 ± 0.08 | 2.18 ± 0.09 | 2.03 ± 0.11 | 1.88 ± 0.11 |
| DiffFoley[25] | 1.98 ± 0.10 | 2.01 ± 0.10 | 2.03 ± 0.13 | 1.97 ± 0.14 |
| IM2WAV[29] | 2.17 ± 0.07 | 2.27 ± 0.09 | 2.29 ± 0.11 | 2.01 ± 0.10 |
| TiVA | **2.74 ± 0.11** | **3.12 ± 0.12** | **2.81 ± 0.12** | **2.47 ± 0.13** |

that TiVA can generate more semantically similar sound according to the shape and texture of bright areas in the Mel-spectrogram. For example, in Case 3, TiVA generates a 'triangular' bright area in the Mel-spectrogram that closely resemble the ground-truth, in contrast to the 'rectangular' pattern produced by DiffFoley. There are similar vertical bars in TiVA generated and ground-truth audio, whereas the baseline generates horizontal bars. In addition, we observe that although the predicted audio layouts are coarse in resolution, they contains important temporal, shape and texture information, contributing to the enhanced generation quality and temporal synchronization of our proposed method.

## 4.3 Experiments on Audio Layout

We investigate three audio layout representations: (1) Onset signals, (2) ODF curves, and (3) our proposed low-resolution Mel-spectrogram, all derived from the ground-truth audio. Utilizing the ground-truth audio layouts in training establishes a theoretical generation performance upper-bound for each representation. Generators are trained for each form, integrating the audio layout with the latent space as outlined in Section 3.2.2, with 1D formats onset and ODF elevated to 2D via an additional convolutional layer; all other training parameters keep the same. Results are presented in Table 3. Our low-resolution Mel-spectrogram representation,

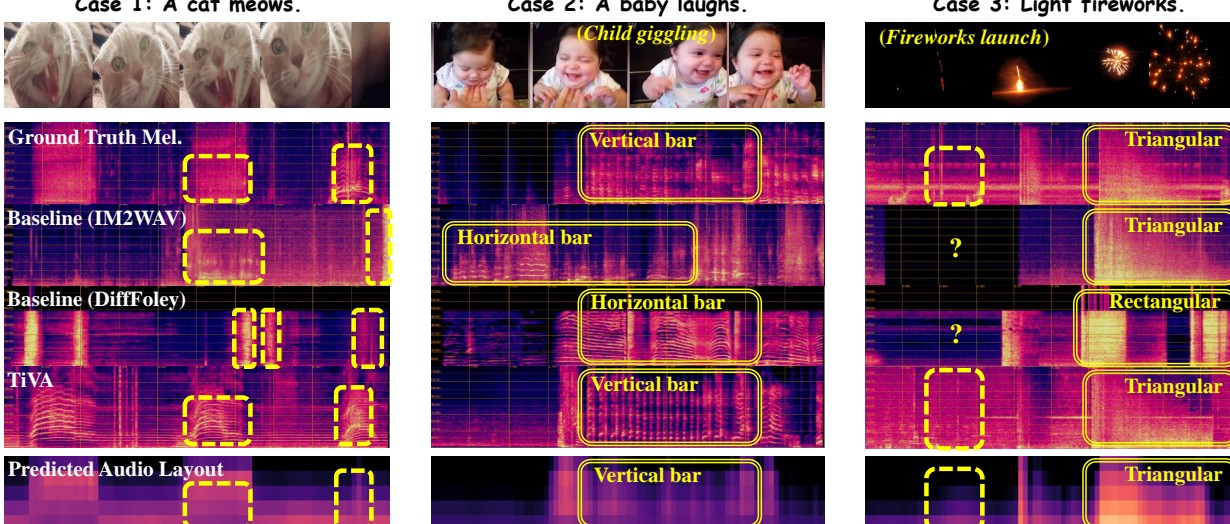

**Figure 4: Case studies on VGGSound test set. The first two rows depict the input video frames and their corresponding ground-truth Mel-spectrograms. The 3rd, 4th and 5th rows show the Mel-spectrograms generated by the baseline methods IM2WAV, DiffFoley, and our proposed method TiVA. The final row illustrates the audio layout produced by TiVA, used for the 5th-row results generation. Each Mel-spectrogram, with time and frequency represented on the horizontal and vertical axes respectively, facilitates visual assessment of temporal and sound structure alignment, by comparing the consistency with the ground-truth.**

**Table 3: Comparison of different audio layout representations. Note: FAD, KL, and IS reported herein are evaluated through their Melception variants.**

| Layout. Reps. | FAD ↓ | KL ↓ | IS ↑ | W. ↓ | DTW. ↓ | IB ↑ |
|---|---|---|---|---|---|---|
| $\mathcal{G}$ w/ Onset | 11.5 | 6.28 | **75.4** | 4.49 | 2.72 | 30.6 |
| $\mathcal{G}$ w/ ODF | 11.7 | 6.26 | 74.5 | 4.60 | 2.68 | **30.7** |
| $\mathcal{G}$ w/ $t_a$ | **8.27** | **3.49** | 56.4 | **2.96** | **2.29** | 29.01 |

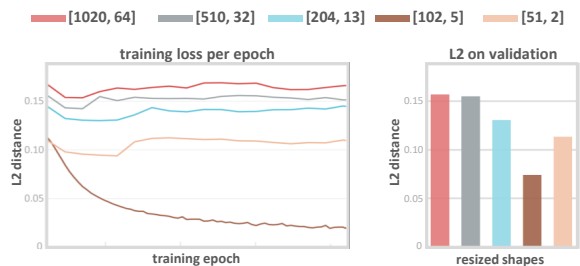

**Figure 5: Training process and performance of audio layout predictor $\mathcal{T}$ across various down-sampling ratios when extracting audio layouts. The [1020, 64], [510, 32], etc., represent the size for down-sampling, corresponding to the hyperparameters $M$ and $C$ in Equation 1. Validation scores are calculated on 200 random samples from the validation set.**

denoted as $t_a$, performs the best across the major metrics FAD, KL, W-dis and DTW-dis. The lower IS score for $t_a$ may attribute to this metric is based on entropy and a more precise control may reduce the diversity of generated audios.

We further explore the influence of down-sampling ratios (resulting in different resolutions) of our proposed audio layouts. We assess the impact of these ratios by examining the performance of predictors with various audio layout resolutions as learning targets. As shown in Figure 5, the training loss curves and L2 loss on the validation set reveal that the compressed size [102, 5] ensures effective learning and precise audio layout prediction; whereas, higher and lower down-sampling ratios increase training losses and hinder model convergence and get worse L2 losses on the validation set.

## 4.4 Experiments on Architecture

Predicting audio layout from visual information serves as the core step in TiVA. We conduct experiments to investigate the optimal design choice for our audio layout predictor: (1) *CLIP features*: the effectiveness of top-layer versus multi-layer CLS tokens from CLIP in Equation 4; (2) *self-attention*: the benefits of windowed and global

self-attention mechanisms; (3) *cross-attention*: the contribution of incorporating detailed patch-level features into the cross-attention. Variants of audio layout predictor are trained on AudioSet-V2A and VGGSound training set, and then validated on VGGSound validation set using L2 distance between predicted and ground-truth audio layouts. Results are listed in Table 4.

Based on the results, we have discerned the following insights: (1) *Multi-layer CLIP features enhance performance*, with three-layer CLS features reducing the L2 distance from 0.208 to 0.187 compared to single-layer CLS features. As it is unclear which visual features are the most important, we retain three-layer CLS features and allow the model to learn on its own; (2) *Self-attention contributes*

**Table 4: Results of different design choices for audio layout predictor $\mathcal{T}$. Each row represents the setting of a variant of the audio layout predictor. CLS[4,8,11] and Patch[4,8,11] refer to the frame features from the CLIP's CLS token and the patch token of the 4-, 8-, 11th layers. 'SA' denotes self-attention layers and 'WA' denotes windowed self-attention layers. L2($t_a, t_v$) denotes the L2 distance of predited audio layouts $t_v$ and ground truth audio layouts $t_a$ on validation.**

| CLIP-features | self-attention | cross-attention | L2($t_a, t_v$) ↓ |
|---|---|---|---|
| CLS[11] | SA | - | 0.208 |
| CLS[4,8,11] | SA. | - | 0.187 |
| CLS[4,8,11] | WA | - | 0.214 |
| CLS[4,8,11] | SA+WA | - | 0.182 |
| CLS[4,8,11] | SA+WA | Patch[4,8,11] | **0.171** |

**Table 5: Ablation study results of different training strategies for TiVA. Note: FAD, KL, and IS reported here are evaluated based on their Melception classifier variants.**

| Ablation | FAD ↓ | KL ↓ | IS ↑ | W. ↓ | DTW. ↓ | IB ↑ |
|---|---|---|---|---|---|---|
| TiVA | **8.71** | **5.98** | 64.9 | **3.85** | 2.59 | **31.0** |
| w/o $\mathcal{T}$ | 9.13 | 6.16 | **79.9** | 4.00 | 2.67 | 30.9 |
| w/o $\mathcal{S}$ | 11.5 | 6.57 | 47.1 | 4.17 | **2.49** | 23.3 |
| w/o Joint | 92.5 | 9.22 | 1.10 | 12.2 | 4.57 | 0.40 |

*more while windowed self-attention is complementary.* Although windowed self-attention alone performs much worse than self-attention alone, an extra gain is achieved when using both. This may be because some audio is delayed, such as a thunder lagging behind lightning, and some audio is inherently long, requiring long-range interactions for accurate prediction; (3) *Incorporating patch features is helpful*, taht further decreases the L2 distance from 0.182 to 0.171. These findings guide our final layout predictor's design.

### 4.5 Experiments on Training Strategies

We further conduct ablation studies on TiVA's training strategies: (1) w/o $\mathcal{T}$, in which we remove audio layout predictor and retrain our framework only with semantic encoder; (2) w/o $\mathcal{S}$, in which we set classifier guidance scale with zero to remove semantic information during generation; (3) w/o Joint, which skips the training of joint optimizing parameters of semantic encoder, audio layout predictor, and generator. We compare them with our full version TiVA in automatic metrics and present results in Table 5. The results indicate that the performance drops the most if without joint training. Compared to w/o $\mathcal{T}$, w/o $\mathcal{S}$ drops more in terms of FAD, KL, W-dis, and IB. This indicates the critical role of semantic alignment in V2A generation, establishing it as a key component even when temporal alignment is controlled. In contrast, w/o $\mathcal{T}$ results in a significant increase in IS, indicating increased diversity of generated audio. This aligns with the intuition that temporal control may reduce diversity to achieve more tightly synchronized audio.

Case 1: A cloud appears and spells "SORA".

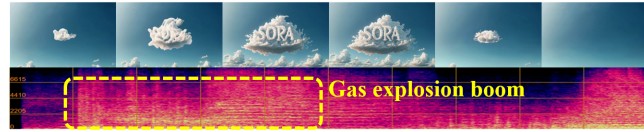

Case 2: A video of a fusion of a lizard and a peacock.

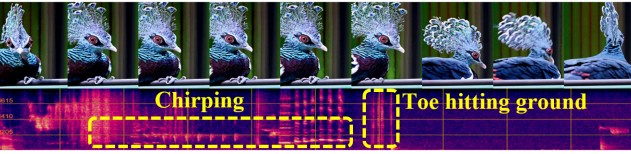

Case 3: A car drives and then a tiger runs to keep up.

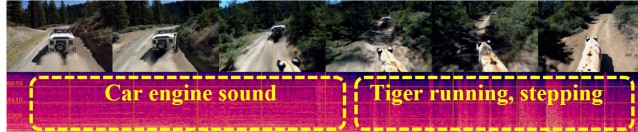

Case 4: In a historic hall, a giant wave crests. Two surfers ride it.

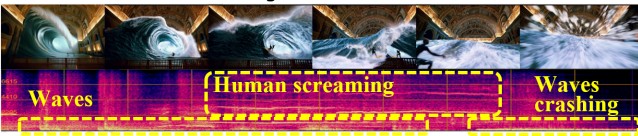

**Figure 6: Some Sora-generated videos with TiVA-produced audios, covering sound effects, fictional creatures, complex transitions, and multiple sound sources in Cases 1-4.**

### 4.6 Case Study of Sora Videos Test

An important application of video-to-audio generation is providing soundtracks for silent videos, including those created by AI. We try to produce audios for videos generated by Sora [4] for testing our TiVA model's generalization capabilities to out-of-domain data. Due to the absence of ground-truth audio, we add our generated audio tracks to Sora-generated videos and show some cases in Figure 6. These cases cover the videos with creative scenes requiring non-natural sound effects (Case 1), fictional creatures (Case 2), complex transitions (Case 3), and scenes with multiple sound sources (Case 4). Our model TiVA successfully generates temporally aligned and semantically coherent audio for these videos, demonstrating promising generalization capabilities to open domains and the feasibility of dubbing sound effects for videos.

### 5 Conclusion and Future Work

This paper introduces TiVA, a novel framework for video-to-audio generation that excels in creating semantically accurate and temporally synchronized audio tracks, with the innovative concept of *audio layout*. Results in objective and subjective evaluations consistently indicate TiVA's generation superiority in quality, semantic relevance, and synchronization. TiVA also shows promising results even for out-of-domain videos, e.g., Sora-generated videos. In future work, we aim to explore audio layouts with alternative generative frameworks and various tasks, and investigate TiVA's scalability.

## Acknowledgments

This work is supported by the National Natural Science Foundation of China (No. 62276268) and ZHI-TECH GROUP.

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
