# OpenReview forum: "TiVA: Time-Aligned Video-to-Audio Generation"
_acmmm.org/ACMMM/2024/Conference — MM2024 Oral_

### Official Review · Reviewer_ifFe · 2024-05-03

**Rating:** 5
**Confidence:** 3

**Summary:**

For the video-to-audio (V2A) generation task, this paper proposes a time-aligned V2A framework TiVA to jointly achieve semantic matching and temporal synchronization. In addition to typical video frames, the method also utilizes low-resolution Mel-spectrograms of audio ground truths (audio layout called in this paper) as an important condition for the diffusion-based model. Experimental results demonstrate the effectiveness of the proposed method.

**Strengths:**

I do believe this is a nice work.
1. The paper is well-written with clear explanations of the proposed method. Figures and tables are effectively used, enhancing understanding of the TiVA framework.
2. The proposed method is simple yet highly effective.
3. Extensive experimental results, including ablation studies, robustly support the validity of the proposed method.

**Limitations:**

I found no major problems with this paper. I just have a few small suggestions or questions.
1. It would be better to also provide citations in Table 1. The meaning of the `CFG' in Table 1 is not provided.
2.  Utilizing the audio mel-spectrogram in the audio generation task may not be entirely novel.  A more exhaustive review of related literature might be beneficial to better position this work within existing research.
3.  Making the source code available would significantly enhance the paper's impact, allowing for reproducibility and further research based on this work.

**Suitability:**

3

---

### Official Review · Reviewer_Dc88 · 2024-05-19

**Rating:** 4
**Confidence:** 3

**Summary:**

In this study, the authors propose a new approach to video-to-audio generation called TiVA. The goal is to generate high-quality audio for silent videos with semantic similarity and accurate temporal synchronization. In particular, TiVA encodes the visual semantics and predicts an audio layout separately to achieve joint semantic matching and precise temporal synchronization.

**Strengths:**

1、The practice of ensuring the consistency of the timing of audio and video generation by first generating the audio layout has certain innovativeness.

2、The authors provide many playable samples in the provided link, which demonstrate the effectiveness of the proposed method.

**Limitations:**

1、Previous methods have used attention mechanisms in addition to semantic guidance for temporal alignment of audio and video, such as MM_Diffusion [1]. However, such alignment in this method relies solely on the low-resolution audio layout. I suspect that the downsampled audio layout may not provide sufficient temporal guidance for multi-channel audio (such as music for dancing or movie soundtracks) or natural sound (such as the sound of ocean waves) because the downsampling operation loses a lot of the fine details of the mel-spectrogram.

2、In  L402, the author states that they used a Transformer-based UNet. What is the difference between this structure and the UNet used in traditional diffusion models? In addition, in Equ. 2 and Equ. 3, the author used various subscripts and superscripts for "s," but did not explain the meaning of each subscript and superscript.

3、The model proposed in this paper needs to be trained separately and then jointly. Why can't it be trained end-to-end directly?

**Suitability:**

3

---

### Official Review · Reviewer_ctLf · 2024-05-23

**Rating:** 4
**Confidence:** 4

**Summary:**

This article proposes a TIVA framework, which aims to generate high -quality audio to the silent video by achieving semantic similarity and accurate time synchronization.
Unlike the existing method that mainly focuses on the method of matching vision and Audio Semantic with the rough time, Tiva encodes Visual Semantic and predicts Audio Layout respectively.
It then uses these components as Condition Signal to train Latent Diffusion Model.
Through comprehensive objective and subjective experiments, Tiva shows excellent performance in Semantic Matching and Temporal Synchronization, exceeding the current most advanced method.

**Strengths:**

1. The starting point of this article is good. To solve the problem of fine-grained alignment in the Audio generation task, it is indeed a interesting and needed research direction in the current field. And from the results provided in this article, the proposed practices are effective in the process of solving this task.

**Limitations:**

1. I questioned some motivation to use Audio Layout. As the author said, he is just the result of Mel-Spectrogram after sampling, so what exactly is this Audio Representation? Why can I include Temporacture after sampling.
2. Based on the above problems, the author needs to provide an additional experiment: Under the settings of Audio Layout, using Audio Vae and Vocoder to reconstruct the original information, it can maintain the original information. Compared with the reconstruction based on the original Mel-Spectrogram.
3. When processing long videos, how to ensure that all video frames in long videos are encoded, and is there any sampling strategies for some frames? The author should provide more details, instead of simply describing the use of embedding with CLIP.
4. The focus of this article is Time-Aligned Video-to-Audio Generation. So it is very important to evaluate whether the Audio generated by the generated by Video is indeed aligned with Video. Therefore, the author should emphasize the evaluation indicators used in detail, including its principles, and how to measure alignment.
5. Formula (6) is defective. "LDM" contains noise sampling, noise estimation, and loss functions, which are not simply used to estimate noise. It is better to write a specific symbol of Denoising Network.

**Suitability:**

3

---

### Official Review · Reviewer_161a · 2024-05-24

**Rating:** 4
**Confidence:** 3

**Summary:**

The paper proposes a new time-aligned video-to-audio generation framework called TiVA, which aims to provide precise temporal synchronization. TiVA uses a unique audio layout to predict the audio based on visual semantics and temporal information, and leverages a ldm for generaton. The results from objective and subjective experiments demonstrate that TiVA outperforms state-of-the-art methods in terms of semantic matching and temporal synchronization. Additionally, TiVA shows promising generalization capabilities when tested on out-of-domain videos.

**Strengths:**

1. TiVA proposes an interesting solution to capture temporal aspects of video content to generate corresponding audio, resulting in a high degree of synchronization.

2. The use of an audio layout to predict audio based on visual information is new.

3. TiVA shows robust performance even on out-of-domain videos, demonstrating its ability to generalize beyond the training data.

**Limitations:**

1. The comparison and discussion with existing approaches could be more in-depth. First, there are some related works missing, e.g,

Seeing and Hearing: Open-domain Visual-Audio Generation with Diffusion Latent Aligners (https://arxiv.org/pdf/2402.17723)

which also shows an interesting demo on generating audio for Sora video. I believe a discussion/comparison is needed here. Also, the comparions with existing approaches might not be fully fair. For DiffFoley, it is designed to generation audio for video that have a strong rythm, like drum beating, water flowing, which requires high-resolution audio-video synchronization capability. And in this paper, the examples shown are more related to some long-lasting sounds, like cat meowing and baby laughs. I think that's why a low-resolution audio layout can work well in this case. The authors might need to provide more justisfications for such cases.

2. Presentation could be improved. In fig3(a), t_a is the input to the network. But during inference, it should be t_v, as there is no GT t_a. Similarly, when introduing t_a and t_v in the paper, it could be strengthened more on the relations between them.

3. In Figure 5, what's the reason behind the gap between [102, 5] and other groups of parameters?

**Suitability:**

3

---

### Meta-Review · Area_Chair_yxn7 · 2024-07-03

**Recommendation:** Accept (Oral)
**Confidence:** 5

**Metareview:**

The paper proposes a new method for time-aligned video-to-audio generation. The paper received 3 WAs and 1 BA. The reviewers all liked the video-to-text problem and agreed that the proposed method is innovative and the consideration of predicting an audio layout is effective. The AC agrees with the reviewers that the video-to-audio work is timely considering the current trend in visual generative models (many on text to image and text to video, but few on video to audio) and marks a good contribution to the field.